Stress adaptive plasticity from Aegilops tauschii introgression lines improves drought and heat stress tolerance in bread wheat (Triticum aestivum L.)

Gudi Santosh 1
Jain Mohit 1
Singh Satinder 1
Kaur Satinder 2
Srivastava Puja 1
Mavi Gurvinder Singh 1
Chhuneja Parveen 2
Sohu Virinder Singh 1
Safhi Fatmah A. 3
El-Moneim Diaa Abd 4
Sharma Achla achla12@gmail.com 1
1 Department of Plant Breeding and Genetics, Punjab Agricultural University , Ludhiana , Punjab , India
2 School of Agricultural Biotechnology, Punjab Agricultural University , Ludhiana , Punjab , India
3 Department of Biology, College of Science, Princess Nourah bint Abdulrahman University , Riyadh , Saudi Arabia
4 Department of Plant Production (Genetic Branch), Faculty of Environmental Agricultural Sciences, Arish University , El-Arish , Egypt
Kutlu Imren
Electronic publication date: 2024 Jun 11
Publication date: 2024
Volume: 12
Electronic Location ID: e17528
Received 2024 Feb 20; Accepted 2024 May 17
Copyright: ©2024 Gudi et al.
Copyright year: 2024
Copyright holder: Gudi et al.
License: This is an open access article distributed under the terms of the Creative Commons Attribution License, which permits unrestricted use, distribution, reproduction and adaptation in any medium and for any purpose provided that it is properly attributed. For attribution, the original author(s), title, publication source (PeerJ) and either DOI or URL of the article must be cited.
License URL: https://creativecommons.org/licenses/by/4.0/

Keywords: Wheat, Abiotic stress, Stress resiliency, Germplasm evaluation, Stability analysis

Funding: The “Transforming India’s Green Revolution by Research and Empowerment for Sustainable food Supplies (TIGR2ESS)” funded by United Kingdom through UK Research and Innovation (UKRI), CGRF (BBSRC) BB/P027970/1 Princess Nourah bint Abdulrahman University, Riyadh, Saudi Arabia PNURSP2024R318 This work was supported by the “Transforming India’s Green Revolution by Research and Empowerment for Sustainable food Supplies (TIGR2ESS)” funded by the United Kingdom through the UK Research and Innovation (UKRI), CGRF (BBSRC), grant number: BB/P027970/1 and the “Princess Nourah bint Abdulrahman University Researchers Supporting Project number (PNURSP2024R318)” Princess Nourah bint Abdulrahman University, Riyadh, Saudi Arabia. Princess Nourah bint Abdulrahman University also assisted with the data analysis. The funders had no role in study design, data collection and analysis, decision to publish, or preparation of the manuscript.

==============================
Aegilops tauchii is a D-genome donor of hexaploid wheat and is a potential source of genes for various biotic and abiotic stresses including heat and drought. In the present study, we used multi-stage evaluation technique to understand the effects of heat and drought stresses on Ae. tauschii derived introgression lines (ILs). Preliminary evaluation (during stage-I) of 369 ILs for various agronomic traits identified 59 agronomically superior ILs. In the second stage (stage-II), selected ILs (i.e., 59 ILs) were evaluated for seedling heat (at 30 °C and 35 °C) and drought (at 20% poly-ethylene glycol; PEG) stress tolerance under growth chambers (stage-II). Heat and drought stress significantly reduced the seedling vigour by 59.29 and 60.37 percent, respectively. Genotype × treatment interaction analysis for seedling vigour stress tolerance index (STI) identified IL-50, IL-56, and IL-68 as high-performing ILs under heat stress and IL-42 and IL-44 as high-performing ILs under drought stress. It also revealed IL-44 and IL-50 as the stable ILs under heat and drought stresses. Furthermore, in the third stage (stage-III), selected ILs were evaluated for heat and drought stress tolerance under field condition over two cropping seasons (viz., 2020–21 and 2021–22), which significantly reduced the grain yield by 72.79 and 48.70 percent, respectively. Stability analysis was performed to identify IL-47, IL-51, and IL-259 as the most stable ILs in stage-III. Tolerant ILs with specific and wider adaptability identified in this study can serve as the potential resources to understand the genetic basis of heat and drought stress tolerance in wheat and they can also be utilized in developing high-yielding wheat cultivars with enhanced heat and drought stress tolerance.

Introduction

Wheat (Triticum aestivum L.) is the major cereal crop in the world, with an annual production of around 808.44 million tons (FAO, 2023). India is the second-largest producer of wheat in the world, with major contributions coming from the northern India, especially from the Punjab and Haryana. November to April is considered as the optimum growing period for wheat in this region. Furthermore, depending on the availability of time for land preparation and the type of rice varieties grown (modern or traditional varieties), farmers in this region will sow wheat either in the last week of October (i.e., early sowing) or mid to late December (i.e., late sowing) (Punjab Agricultural University, 2023; Gudi et al., 2022a). Early sowing under residual moisture conditions will expose wheat crop to heat and drought stress at the seedling stage, which reduces the germination percentage and seedling vigour. However, delayed sowing (i.e., from December to January) owing to late harvesting of traditional rice varieties or other long duration and area specific crops like potato, sugarcane, turmeric, cotton etc., will expose the wheat to terminal heat stress. This will reduce the pollen viability and grain filling duration, thereby affecting the grain yield (Punjab Agricultural University, 2023). Furthermore, the fluctuating rainfall due to frequent heat waves, depleting water table and scarcity of electricity (to irrigate wheat fields) have a substantial impact on wheat production in this region (Singh, 2012; Baweja et al., 2017). For instance, the severity and spread of heat wave 2022 and 2023 in the northern and central India, coupled with no rainfall during March and April, witnessed its impact on wheat production (with a predicted yield loss of 4.41%) (Bal, Prasad & Singh, 2022).

Heat and drought stress can occur either independently or together, and in either case, they significantly reduce wheat productivity. For instance, about 69 percent of the crop yield losses were attributable to the combined effects of heat and drought (Boyer, 1982; Mao et al., 2023; Gudi et al., 2024). These stresses reduces grain number, grain weight, grain yield, and grain quality by limiting the synthesis and translocation of photosynthetic assimilates to the developing grains (Dwivedi et al., 2017; Mao et al., 2022; Tanin et al., 2023). Heat stress during anthesis and grain filling stages, respectively, reduces the wheat yield by 302 and 161 kg/ha/°C for each day with a maximum temperature in excess of 30 °C (Telfer et al., 2018). Heat stress from anthesis to maturity causes yellowing and shriveling of grains and also reduces grain yields by up to 15–25 percent (Bal, Prasad & Singh, 2022; Gudi et al., 2023). However, drought stress reduces wheat yield, with an average yield loss of 17–70 percent (Nouri-Ganbalani, Nouri-Ganbalani & Hassanpanah, 2009). Furthermore, due to erratic temperatures and rainfall, about 75 percent of wheat-growing countries experience a decrease in annual wheat production of 5.5 percent (Lobell, Schlenker & Costa-Roberts, 2011). Therefore, there is an urgent call for developing high-yielding wheat cultivars that can thrive better under heat and drought stresses.

The narrow genetic variation present in the wheat gene pool limits the development of high-yielding, stress resilient wheat cultivars (Bailey-Serres et al., 2019; Singh et al., 2022). Therefore, it is imperative to identify and exploit the wild germplasm resources to enhance the genetic diversity and to deliver heat and drought responsive wheat varieties. The Aegilops tauschii, a diploid progenitor of wheat, is an important source of genes for several abiotic stresses, including heat and drought (Waines, 1994; Valkoun, 2001; Khan et al., 2017; Gudi et al., 2022b; Li et al., 2024). The Ae. tauschii derived introgression lines (ILs) shows variable tolerance to several abiotic stresses (van Ginkel & Ogbonnaya, 2007; Li et al., 2024). For instance, field evaluation of Ae. tauschii derived ILs identified the genomic regions associated with heat tolerance and increased the grain yield under heat stress (Molero et al., 2023). Such ILs carrying favorable genomic regions could be used as pre-breeding material to improve stress tolerance in wheat.

With these objectives, the wheat breeding group at Punjab Agricultural University, Ludhiana, screened large number of Ae. tauschii accessions (nearly 250 accessions) to identify heat and drought tolerant accessions (Kaur et al., 2021). Drought and heat tolerant Ae. tauschii accessions were utilized in developing synthetic hexaploids and Ae. tauschii derived ILs (i.e., 369 BC1F6/BC1F7 ILs). In the present study, we carried out the multi-stage evaluation of Ae. tauschii derived ILs. Stage-I include the evaluation of 369 ILs under field condition for various agronomic traits. Furthermore, the best performing ILs from stage-I were evaluated for studying the effect of heat and drought stress under growth chamber (stage-II) and field condition (stage-II). Finally, the stability analysis was performed to select the stable and high-yielding ILs under heat and drought stress.

Materials and methods

The detailed materials and methods used during study at each of these stages have been discussed below-

Stage I: preliminary evaluation of Ae. tauschii derived ILs for agronomic traits

Total 382 germplasm lines including 369 BC1F6/BC1F7 ILs, 11 parental lines used in developing these ILs, and two commercial checks (viz., PBW725 and HD3086), were evaluated for various agronomic traits. Of the 11 parental lines, two were T. durum wheat varieties (viz., PBW114 and PDW233), six were synthetics (viz., Syn1 to Syn 6), and three were advanced breeding lines (ABLs) (viz., BWL3279, BWL3531, and BWL4444) (Table S1). All the germplasm lines have been developed and are being maintained at the Department of Plant Breeding and Genetics, Punjab Agricultural University, Ludhiana. The details of the crossing procedure and parental lines used in developing these ILs are shown in Fig. 1.

Figure 1 Development of Aegilops tauschii introgression lines (ILs) and their evaluation for heat and drought stress tolerance.

The figure includes two parts. The first part includes the procedure for developing Ae. tauschii derived ILs by via synthetic hexaploid line production. The second part includes the evaluation of ILs at three different stages. In stage-I, 369 ILs were evaluated for agronomic traits and 59 agronomically superior lines were selected. In stage-II, 59 agronomically superior ILs were evaluated for seedling heat and drought stress tolerance in the growth chambers. In stage-III, 59 ILs were evaluated for heat and drought stress tolerance under field condition over two cropping seasons (viz., 2020–21 and 2021–22).

All germplasm lines were sown in the augmented block design during 2019–20 cropping season. The experimental field was divided into 10 blocks with 40 plots (which includes 38 germplasm lines and two check varieties) in each block. Therefore, the experiment included a total of 400 plots. Each plot consisted of four rows of one meter length, with a row spacing of 20 cm, and the distance between each plot was 50 cm. The sowing was done during the first week of November, which is the ideal sowing time for wheat in the Punjab, India.

The agronomic data on spike length (cm), spikelet per spike, plant height (cm), and tiller number per meter length were recorded. Furthermore, visual observations were also made on days to anthesis, days to maturity, and stay-green type phenotype during growing cycle. Of the 369, 59 ILs with good agronomic performance (such as stay green, non-lodging, and having optimum days to flowering and maturity) were selected and harvested to measure grain yield (gm/plot) and 1,000 seed weight (gm). In addition, of the total 11 parental lines, only nine parental lines contributing to these 59 ILs (except PDW233 and Syn6) were also harvested.

Stage II: Evaluation of selected ILs for heat and drought stress tolerance under growth chambers

The experimental material comprised 74 germplasm lines, including 59 ILs (selected from stage-I), nine parental lines, and six check varieties (viz., PBW343, PBW660, C306, Raj3765, PBW725, and HD3086) (Fig. 1; Table S2). The experiment was laid out in a completely randomized design (CRD) with three replications and four treatments under controlled conditions. The details of the treatments used include: T1-control (grown in distilled water at 20 °C temperature); T2-mild heat stress (grown in distilled water at 30 °C temperature); T3-severe heat stress (grown in distilled water at 35 °C temperature); and T4-drought stress (grown in 20% poly-ethylene glycol (PEG) at 20 °C temperature).

Seedling characteristics were accessed using the modified cigar roll method of seed germination (Zhu, Kaeppler & Lynch, 2005). Bold and uniform seeds from each genotype were selected and disinfected with 0.1% HgCl2 for 20–30 min and 70% ethanol for 10–15 min. Later seeds were thoroughly washed with deionized water (diH2O) for three times to remove the HgCl2 and ethanol. Twenty seeds from each germplasm lines were placed horizontally on germination paper moistened with distilled water (for T1, T2, and T3) and 20% PEG solution (for T4; to ensure drought stress from sowing itself). Subsequently, germination papers were rolled and were shifted immediately to respective growth chambers (adjusted to three temperature regimes; 20 °C, 30 °C, and 35 °C) to ensure the heat stress from the beginning. Then seeds were allowed to germinate under dark conditions for first three days. Once seeds were germinated, the growth chambers were adjusted to a 16 h photoperiod (i.e., 16 h of light and 8 h of darkness) at three temperature regimes (i.e., 20 °C, 30 °C, and 35 °C) for the next 12 days. On the 12th day after germination, data on germination percentage, shoot length (cm), and root length (cm) were recorded. The seedling vigour was calculated by using the following formula: Seedling vigour=shoot lengthcm+root lengthcm×germination percentage%100

The stress tolerance index (STI) for seedling vigour under heat (T2 and T3) and drought (T4) stresses was derived as the ratio of seedling vigour under stress to the seedling vigour under controlled conditions, multiplied by 100. Furthermore, based on seedling vigour STI, the ILs were classified as tolerant (i.e., STI ≥ 80%), moderately tolerant (i.e., STI = 50–80%), and susceptible (i.e., STI ≤ 50%).

Stage III: Field evaluation of selected ILs for heat and drought stress tolerance

Germplasm lines used in stage-II (viz., 74 lines) were also evaluated for heat and drought tolerance under field conditions for two consecutive years (viz., 2020–21 and 2021–22) (Fig. 1; Table S2). During each cropping season, plant material was sown under four different environmental conditions viz. early (early heat stress), timely sowing (control), late sowing (terminal heat stress), and drought conditions. Details of field environments used for assessing heat and drought stress tolerance are depicted in Table 1.

Table 1 Different environments used to screen introgression lines (ILs) for heat and drought stress tolerance during stage-III.

Environments	Sowing date	Environmental conditions	Soil temperature (in °C; at 5 cm depth)	
E1	27th October, 2020	Early sowing (Early heat stress)	25.3	
E2	25th October, 2021	22.8	
E3	28th November, 2020	Timely sowing (Control)	17.8	
E4	3rd December, 2021	20.2	
E5	15th December, 2020	Late sowing (Terminal heat stress)	16.8	
E6	25th December, 2021	14	
E7	28th November, 2020	Rainfed (Drought)	17.8	
E8	3rd December, 2021	20.2	

The experiment was laid out in a randomized complete block design (RCBD) with two replications in all eight environments. The plots were 1.6 m2 in size and separated by 50 cm. Each plot included four rows of two-meters length, with a row spacing of 20 cm. Data on days to 50% heading, spikelet number per spike, spike length (cm), plant height (cm), tiller number per meter, days to maturity, and grain yield per plot (gm) were collected.

Experimental area, agronomic practices, and weather data

Field and lab experiments were conducted in the experimental area and wheat laboratory of the Department of Plant Breeding and Genetics, Punjab Agricultural University, Ludhiana (30.9° North and 75.86° East, with a mean sea level of 244 m). The predominant soil type at the experimental site was a sandy loam with a slightly alkaline condition (pH from 7.8 to 8.5). The climate of the experimental location was sub-tropical and semi-arid with daily minimum temperatures ranging from 0 to 4 °C during December-January and the maximum temperature ranging from 39 to 45 °C during May (Sharma et al., 2021). Land preparation, including plowing, harrowing, and flanking, was done to achieve fine tilth, and the crop was raised using standard agronomic practices suggested for the Punjab region. To ensure sufficient soil moisture, the field was irrigated with a total of four irrigations (with 75 mm per irrigation) during stage-I and stage-III (for the first six environments; viz., E1–E6). However, last two environments (viz., E7 and E8) in stage-III were grown as rainfed wheat to ensure drought stress. Pre-emergence (pendimethalin; Stomp at 1.5 liter per acre) and post-emergence (weed and time-specific) herbicides were used to control the weeds. Prevailing insects, such as aphids, were controlled by spraying thiamethoxam (Actara at 20 gm per acre). Crop was harvested when it reached physiological maturity and agronomic data were collected on regular basis. Furthermore, the weekly weather data on mean maximum and minimum temperatures (°C), total rainfall (mm), mean sun-shine hours (hours/day), and mean relative humidity (%) were accessed from the Department of Climate Change and Agricultural Meteorology, Punjab Agricultural University, Ludhiana (Fig. S1).

Statistical analyses

Analysis of variance (ANOVA) for augmented block design (in stage-I), CRD (in stage-II), and RCBD (in stage-III) were done in Agricolae package built in the RStudio (R Core Team, 2020). Pearson’s correlation coefficient analysis, linear regression analysis, genotypic main effect plus genotype × environment interaction (GGE) biplot, additive main effects and multiplicative interaction (AMMI), and weighted average of absolute scores for the best linear unbiased predictions (BLUPs) of the genotype × environment interaction (WAASB) analysis were made using “Metan package” built in RStudio.

Results

Stage I: Preliminary evaluation of ILs for agronomic traits

Data on spike length, spikelet number, plant height, and tiller number were collected from all the germplasm lines, including ILs (369), parents (11), and check verities (2). Analysis of variance (ANOVA) for augmented block design with adjusted mean showed significant differences (P-value < 0.05) among germplasm lines (Table S3). Of the 369 ILs, 59 lines having good agronomic characteristics such as compact spike (i.e., short spikes (7.8–20.7 cm) with a greater number of spikelets per spike(15–25.7) (Figs. 2A, 2B), reduced plant height (i.e., non-lodging; 67.33–114 cm) (Fig. 2C), optimum number of tillers (30–129) (Fig. 2D), optimum days to anthesis and maturity, and stay green type, were selected and harvested. The selected ILs showed significant variations for seed yield (gm per plot), with the highest yielding line, IL-134 (with 675 gm/plot) and the lowest yielding line, IL-279 (150 gm/plot) (Fig. 2E). The selected ILs also showed a significant variation for 1,000 seed weight, which ranged from 24.4 (IL-227) to 52.8 gm (IL-50) (Fig. 2F). Among the selected ILs, highest yielding line i.e., IL-134 (675 gm/plot), also showed higher seed yield over parental lines and both the check varieties i.e., PBW725 (619.35 gm/plot) and HD3086 (527.25 gm/plot). Furthermore, IL-29 (46.2 gm), IL-50 (52.8 gm), IL-42 (45.8 gm), and IL-134 (48.8 gm) showed higher 1,000 seed weight over parents and check varieties.

Figure 2 Comparing the agronomic performance of Aegilops tauschii derived ILs (i.e., 369 ILs) and selected ILs (i.e., 59 only) during stage-I.

(A) Spike length (cm); (B) spikelet number; (C) plant heigh (cm); (D) tiller number; (E) grain yield (gm per plot); (F) 1,000 seed weight (gm).

Pearson’s correlation coefficient analysis, multiple linear regression analysis, and principal component analysis (PCA) based on selected ILs revealed a significant and positive association between seed yield and seed weight (p < 0.001) (Fig. S2). However, spike length showed a significant negative correlation with seed yield (p < 0.05) and seed weight (p < 0.001). In addition, seed weight showed a significant (p < 0.01) negative association with spikelet number and plant height. Moreover, seed yield has not shown a significant association with tiller number, plant height, and spikelet number.

Stage II: Evaluation of selected ILs for seedling heat and drought stress tolerance

ILs selected from the stage-I along with their parental lines and check varieties were subjected to seedling heat (i.e., 30 °C; T2 and 35 °C; T3) and drought stress (i.e, 20% PEG; T4). Heat and drought stresses significantly reduced the germination percentage, shoot length (except under T2), root length, and seedling vigour (Figs. 3A–3D; Table S4). Heat stress reduced the seedling vigour by 21.49 and 59.29 percent under T2 and T3, respectively. Exposure to drought stress also reduced the seedling vigour by 60.37 percent. Furthermore, pooled ANOVA for STI revealed the significant genotype × heat interaction (Table 2). Based on STI, 30 ILs were classified as heat tolerant under T2, three ILs were classified as heat tolerant under T3 (IL-50, IL-56 and IL-68), and two ILs (IL-42 and IL-44) were classified as drought tolerant (T4) (Fig. 4; Table 3). Three ILs that showed tolerance under severe heat stress (i.e., T3) also showed tolerance under T2. Heat (IL-50, IL-56, and IL-68) and drought tolerant ILs (IL-42 and IL-44) outperformed the parental lines and check varieties. Additionally, IL-50, a heat tolerant line also performed moderately under drought stress. In contrast, a drought tolerant IL, IL-44, performed moderately under extreme heat stress (i.e., 35 °C), whereas IL-42 showed high level of tolerance under T2 (i.e., 30 °C) (Table 3).

Figure 3 Effect of heat and drought stress on various seedling characteristics in stage-II.

(A–E) effect of stress treatments on germination percentage (%), root length (cm), shoot length (cm), seedling vigour, and stress tolerance index; (F) Pearson’s correlation coefficient analysis between different stress treatments (T1-T4); (G–L) linear regression analysis for seedling vigour under T1 with T2 (G), T1 with T3 (H); T1 with T4 (I), T2 with T3 (J), T2 with T4 (K), and T3 with T4 (L).

Table 2 Effect of heat and drought stresses on stress tolerance index (STI) of seedling vigour.

Stress	Source of variation	Df	Sum sq	Mean sq	F value	Pr(>F)	
Heat	Replication	2	713	356	–	–	
Genotype	73	124,793	1,709	11.664	<0.001	
Ea	146	21,398	147	–	–	
CV	19.8	
LSD	13.83	
Heat	1	169,725	169,725	2,580.346	<0.001	
Genotype*Heat	73	66,206	907	13.788	<0.001	
Eb	148	9,735	66	–	–	
CV	13.3	
LSD	1.52	
Drought	Replication	2	105	52.44	1.8093	0.1674	
Genotype	73	73,397	1005.44	34.6905	<0.001	
Residuals	146	4,232	28.98	 	 	
CV	13.68	
LSD	8.69	
Notes.

Df degrees of freedom

Ea heat residual

CV coefficient of variation

LSD least significant difference

Eb genotype residual

Figure 4 Phenotypic performance of heat and drought tolerant introgression lines identified based on seedling vigour stress tolerance index (STI) in stage-II.

IL-50, IL-56, and IL-68 were identified as heat stress tolerant ILs, whereas, IL-42 and IL-44 were identified as drought stress tolerant ILs.

Table 3 Characterization of introgression lines (ILs) for drought and heat stress tolerance using stress tolerance index (STI).

Stress	STI (tolerance)	No. of ILs	Introgression lines	
	>80 (tolerant)	30	IL-56, IL-30, IL-91, IL-50, IL-42, IL-226, IL-308, IL-51, IL-66, IL-44, IL-45, IL-87, IL-40, IL-68, IL-11, IL-184, IL-135, IL-14, IL-180, IL-105, IL-185, IL-41, IL-27, IL-103, IL-279, IL-47, IL-227, IL-54, IL-80 and IL-21	
T2: heat stress at 30°C	50–80 (moderate tolerant)	24	IL-29, IL-15, IL-248, IL-79, IL-77, IL-123, IL-136, IL-102, IL-95, IL-159, IL-355, IL-10, IL-39, IL-35, IL-32, IL-115, Syn3, IL-130, Syn5, IL-106, IL-119, IL-134, IL-131, IL-19, IL-303 and IL-186	
	<50 (succeptible)	5	IL-120, IL-98, IL-364, IL-212 and IL-259	
	>80 (tolerant)	3	IL-50, IL-56 and IL-68	
	50–80 (moderate tolerant)	22	IL-136, IL-54, IL-44, IL-47, IL-135, IL-123, IL-35, IL-45, IL-14, IL-227, IL-77, IL-180, IL-105, IL-106, IL-98, IL-30, IL-29, IL-186, IL-185, IL-87, IL-21 and IL-248	
T3: heat stress at 35°C	<50 (succeptible)	34	IL-95, IL-308, IL-226, IL-19, IL-184, IL-27, IL-15, IL-66, IL-279, IL-120, IL-39, IL-11, IL-32, IL-131, IL-103, IL-259, IL-130, IL-212, IL-159, IL-119, IL-10, IL-42, IL-80, IL-102, IL-303, IL-51, IL-115, IL-40, IL-134, IL-91, IL-41, IL-355, IL-79 and IL-364	
	>80 (tolerant)	2	IL-42 and IL-44	
	50–80 (moderate tolerant)	8	IL-115, IL-364, IL-11, IL-50, IL-35, IL-103, IL-47 and IL-21	
T4: drought stress at 20% PEG	<50 (succeptible)	49	IL-54, IL-226, IL-227, IL-135, IL-30, IL-259, IL-29, IL-184, IL-15, IL-32, IL-136, IL-308, IL-40, IL-248, IL-123, IL-355, IL-105, IL-131, IL-45, IL-185, IL-41, IL-303, IL-102, IL-19, IL-10, IL-14, IL-120, IL-212, IL-51, IL-91, IL-27, IL-95, IL-106, IL-159, IL-39, IL-279, IL-186, IL-130, IL-134, IL-98, IL-119, IL-66, IL-80, IL-87, IL-68, IL-79, IL-56, IL-77 and IL-180	

The pearson’s correlation coefficient analysis showed a significant and positive association for the seedling vigour under different treatments except for T3 and T4 (Fig. 3F) (Table 3). In addition, seedling vigour under, (i) T2, T3, and T4 was regressed on the seedling vigour under T1 (control condition) (Figs. 3G–3I); (ii) T3 and T4 were regressed on the seedling vigour under T2 (Figs. 3J, 3K); and (iii) T4 was regressed on the seedling vigour under T3 (Fig. 3L). For every unit increase in seedling vigour under controlled condition, corresponding seedling vigour under T2, T3, and T4 increased by a factor of 0.79, 0.41, and 0.40, respectively. Similarly, for every unit increase in seedling vigour under T2, there was an increase in seedling vigour by a factor of 0.52 and 0.5 under T3 and T4, respectively. However, seedling vigour under T3 has no influence on seedling vigour under T4. Furthermore, genotype × treatment interaction analysis identified the best performing ILs for heat (IL-50, IL-56, and IL-68) and drought (IL-42 and IL-44) stresses (Fig. 5A). It also revealed the IL-44 and IL-50 as the most stable ILs under both heat and drought stresses (Fig. 5B).

Figure 5 GGE-biplot analysis for seedling vigour stress tolerance index (STI).

(A) Which-won-where view of GGE-biplot; (B) mean vs. stability biplot.

Stage III: Field evaluation of selected ILs for heat and drought stress tolerance

Field experiments were carried to evaluate the heat (E1 to E6) and drought (E7 and E8) stress tolerance in the ILs selected from stage-I along with their parents and check varieties over two cropping seasons (viz., 2020–21 and 2021–22). Heat and drought stress significantly (<0.05) reduced all the agronomic traits including grain yield (Fig. S3 and Table S5). Additionally, pooled ANOVA revealed the significant (<0.05) effect of genotype, stress (viz., heat and drought), year, and their interactions on studied traits including grain yield (Table 4). Based on mean data from both years, IL-77 (921 gm/plot) (under timely sowing), IL-41 (307.7 gm/plot) (under late sowing), and five ILs, IL-123 (565 gm/plot), IL-106 (513 gm/plot), IL-68 (486 gm/plot), IL105 (486 gm/plot), and IL-29 (482.5 gm/plot) (under drought stress) showed higher grain yield than parental lines and check varieties (Table 5). The list of high-yielding ILs identified for the specific and combination of environments is given in Table 5.

Table 4 Effect of heat and drought stresses on various agronomic traits over two years (during stage-III).

Source of variation	Df	DTH	SN	SL	PH	TN	Maturity	Yield	
Heat stress	
Replication	1	581***	1.28	0.14	2.00	46.00	2.00	1197.00	
Genotype	73	196***	31.72***	22.66***	705***	1,795***	64***	96,075***	
Heat	2	34,450***	137.66***	215.1***	51,482***	61,653***	156,057***	28,241,631***	
Year	1	1,113***	171.04***	40.4***	5,408***	12,721***	43,849***	7,759,154***	
Genotype*Heat	146	89***	8.42***	3.91***	146***	624***	35***	64,861***	
Genotype*Year	73	2	4.74***	2.67***	143***	2,137***	2	14,005***	
Heat*Year	2	470***	2.58**	2.17**	149***	2,325***	14,909***	2,129,596***	
Genotype*Heat*Year	146	2	1.81***	1.21***	12***	106***	2	11,720***	
Residuals	443	4.00	0.41	0.31	7.00	61.00	3.00	511.00	
Genotype.LSD	1.60	0.51	0.45	2.12	6.27	1.39	18.14	
Heat.LSD	0.32	0.10	0.09	0.43	1.26	0.28	3.65	
Year.LSD	0.26	0.08	0.07	0.35	1.03	0.23	2.98	
Drought stress	
Replication	1	752***	8.4***	0.16	7.00	143.00	2.00	448.00	
Genotype	73	118***	19.7***	10.62***	343***	1,792***	24***	44,522***	
Drought	1	43643***	55.3***	46.16***	62,213***	15,539***	180,286***	41,325,002***	
Year	1	190***	650.6***	12.13***	4,496***	5,155***	16,223***	1,421,490***	
Genotype*Drought	73	70***	10***	5.96***	171***	500***	19***	45,795***	
Genotype*Year	73	70***	9.2***	5.76***	183***	1,606***	7***	49,446***	
Drought*Year	1	115***	149.9***	0.46	220***	41.00	8,851***	57,237***	
Genotype*Drought*Year	73	72***	7.6***	4.78***	147***	227***	8***	50,114***	
Residuals	295	5.00	0.50	0.50	2.00	72.00	3.00	420.00	
Genotype.LSD	2.20	0.70	0.70	1.39	8.35	1.70	20.17	
Drought.LSD	0.36	0.11	0.11	0.23	1.37	0.28	3.32	
Year.LSD	0.36	0.11	0.11	0.23	1.37	0.28	3.32	
Notes.

Signif. codes: 0.001 ‘***’ 0.01 ‘**’ 0.05 ‘*’ (Tukey test of significance).

Df Degrees of freedom

DTH Days to heading

SN Spikelete number

SL Spike length (cm)

PH Plant height (cm)

TN Tiller number

LSD Least significant difference

Table 5 List of introgression lines (ILs) identified for the specific and combination of environments.

Sowing conditions	Tolerant introgression lines	
Early sown	IL-364 and IL-259	
Timely sown	IL-77, IL-30, and IL-259	
Late sowing	IL-41, IL-259, IL30, and IL-29	
Drought	IL-123, IL-106, IL-68, IL105, IL-29, and IL-30	
Early and Timely sown	IL-259	
Early and Late sown	IL-259	
Early sown and drought	IL-186	
Timely and late sown	IL-30 and IL-259	
Timely sown and drought	IL-30 and IL-105	
Late sown and drought	IL-30 and IL-29	
Early, timely, and late sown	IL-259	
Timely, late sown, and drought	IL-30	

ILs showed higher grain yield under early sowing (E1 and E2) (840.57 gm/plot) than in the timely sowing (E3 and E4) (608.53 gm/plot) (Fig. 6). This is owing to the availability of extended grain filling duration in early sowing as it experiences less terminal heat stress during anthesis and grain filling stages (Figs. 6A–6G). In contrast, grain yield under late sowing (228.72 gm/plot) was greatly reduced due to increased temperature at anthesis and grain filling stages (Fig. 6). affected due to increased values of several climatic co-variates (viz., mean maximum temperature, number of days with >30 °C, and number of days with >35 °C) during anthesis and grain filling stages (Fig. 6). For instance, one-degree Celsius increase in the mean growing season temperature (from sowing to maturity), the grain yield was reduced by 583 gm/plot/°C (Fig. 6; Table S6). The mean maximum temperature during the grain filling stage (i.e., flowering to maturity) was more severe (with a yield reduction of 72.72 gm/plot/°C) than that of anthesis (with a yield reduction of 62.84 gm/plot/°C) (Fig. 6; Table S6). In contrast, the number of days with more than 30 °C and 35 °C had the maximum impact during anthesis stage (with a yield reduction of 71.98 and 611.85 gm/plot/day, respectively) than during the grain filling stage (with yield reduction of 50.99 and 53.21 gm/plot/day, respectively). Drought stress also reduces the grain yield by 48.70 percent (with a yield reduction of 9.88 gm/plot/cm of irrigation water) (Fig. 6H).

Figure 6 Effects of climate covariates on grain yield in stage-III.

(A) Weekly mean temperature and rainfall over two cropping seasons (i.e., 2020–21 and 2021–22); (B) effects of seasonal mean temperature on the grain yield; (C) mean maximum temperature during anthesis on the grain yield: (D) number of days with >30 °C during anthesis on the grain yield; (E) mean maximum temperature during grain filling on the grain yield; (F) number of days with >30 °C during grain filling on the grain yield; (G) number of days with >35 °C during grain filling on the grain yield; (H) and total water (irrigation and rainfall) on the grain yield.

We also carried the multiple linear regression analysis to compare the association among the environments (Fig. 7). This revealed a weak positive association among sowing dates (viz., early, timely, and late) except for timely sowing with late sowing (R2 = 0.12; p-value < 0.01) (Fig. 7). Similarly, we observed weak association between sowing dates with drought stress, except for the timely sowing and drought stress (R2 = 0.11; p-value < 0.01) (Fig. 7).

Figure 7 Linear regression analysis for grain yield under heat and drought stresses.

(A) Regression analysis of early sowing with timely sowing; (B) regression analysis of early sowing with late sowing; (C) regression analysis of early sowing with drought; (D) regression analysis of timely sowing with late sowing; (E) regression analysis of timely sowing with drought; (F) regression analysis of late sowing with drought.

Stability analysis

Even though there was a significant genotype × year interaction; the years 2020-21 (E1, E3, E5, and E7) and 2021–22 (E2, E4, E6, and E8) differentiated ILs in a similar manner for grain yield, as depicted by biplots (Fig. 8). The which-won-where view of GGE-biplot (Fig. 8A), AMMI analysis (Fig. 8B), and nominal yield based on WAASB (Fig. 8C) identified specifically adapted ILs for early sowing (E1 and E2), timely sowing (E3 and E4), late sowing (E5 and E6), and drought stress (E7 and E8) (Table 5).

Figure 8 Stability analysis for grain yield in stage-III.

(A) Which-won-where view of GGE-biplot; (B) which-won-where view of AMMI model; (C) comparing interaction principal component analysis (IPCA) and nominal yield based WAASB scores; (D) mean vs. stability biplot for grain yield; (E) grain yield vs. WAASB biplot.

Classifying genotypes based on interaction-PCA1 (IPCA1) (57.76%) have identified IL-98 as the most stable IL (due to smallest IPCA1 score of 0.01) (Fig. 8C; Table S7). The joint interpretation of mean yield vs stability biplot (GGE biplot) based on PC1 (51.44%) and PC2 (27.32%) identified IL-259 as the stable, high-yielding genotype (Fig. 8D). However, of the total, only 57.76 (IPCA1) and 78.76 percent (PCA1 and PCA2) of the genotype × environment interaction (GEI) variation was explained by AMMI and GGE biplots, respectively, whereas the remaining variation was not included by these biplots (Figs. 8C, 8D). Therefore, to include the maximum GEI variation, WAASB scores were derived by retaining different number of IPCAs (Fig. S4). Plotting grain yield vs. WAASB scores identified IL-98 as the most stable genotype (due to lowest WAASB score i.e., 0.16) (Fig. 8E; Table S6). Whereas, IL-30, IL-259, and IL-364, were considered as high-yielding genotypes with less stability (Fig. 8E). Even though WAASB includes all significant IPCA (P-value < 0.01) axes, it will not give weightage to stability and yield performance while giving ranks to genotypes. Therefore, to overcome this, WAASBY scores (also known as superiority index) were calculated by assigning different weights to stability (100 to 0) and yield performance (0 to 100) (Fig. S5).

The WAASBY scores were calculated by giving 65% weightage to grain yield and 35% weightage to stability (Fig. 9). This helped in identifying IL-47, IL-51, and IL-259 as the high-performing stable ILs under heat and drought stress.

Figure 9 Ranking of 59 Aegilops tauschii derived introgression lines (ILs) based on the superiority index values (i.e., WAASBY scores).

Comparisons among the stages

Comparing the different screening methods (i.e., stages) based on yield was not possible as stage-II includes only seedling evaluation. Therefore, to compare the efficiency of three stages, we used yield data from stage-I and stage-III (early, timely, late, and drought sown) and seedling vigour from stage-II (T1, T2, T3, and T4). The Pearson’s correlation coefficient analysis revealed a weak association between seedling vigour and grain yield (in stage-I and stage-III), except for seedling vigour under T2 and yield under drought (P value < 0.01) (Table 6). The grain yield under stage-I has a significant positive association with grain yield under timely sown (<0.01) and drought (<0.01) conditions of stage-III.

Discussion

Heat and drought are the major abiotic stresses affecting the various developmental stages of wheat (Lobell et al., 2015; Singh et al., 2023; Gudi et al., 2023; Gudi et al., 2024). Wheat wild relatives act as a reservoirs of stress tolerant genes including heat and drought stress tolerance (Singh et al., 2022; Luo et al., 2017; Gaurav et al., 2022). Diploid progenitor of wheat, Ae. tauschii, have high level of resistance or tolerance to several biotic and abiotic stresses. For instance, Ae. tauschii serve as a source of resistance genes for stem rust (Sr33, Sr45, Sr46, and SrTA1662), leaf rust (Lr42) (Lin et al., 2022; Arora et al., 2019), and stripe rust (Yr28) (Athiyannan et al., 2022). Similarly, several Ae. tauschii accessions has already been characterized as tolerant to heat and drought stresses due to the presence of stay green property, cell membrane thermostability, and high-pollen viability (Hairat & Paramjit, 2015; Kaur et al., 2021; Gaurav et al., 2022; Hasanpour et al., 2023; Abbas et al., 2023). Therefore, several efforts were made to introgress the promising genomic regions associated with heat and drought stress tolerance from Ae. tauschii into wheat (van Ginkel & Ogbonnaya, 2007). Efforts were also made at the Punjab Agricultural University, Ludhiana to develop the Ae. tauschii derived ILs to breed climate resilient wheat cultivars (Kaur et al., 2021). In the present study, we evaluated very large set of Ae. tauschii derived ILs (i.e., 369 ILs) for their agronomic performance during stage-I. The ILs showed a huge variation for studied agronomic traits and the 59 agronomically superior ILs were selected for further evaluation under heat and drought stress.

Table 6 Comparison among the screening methods (i.e., stage-I, stage-II, and stage-III).

Stage	Treatments	Stage-II	Stage-I	Stage-III	
VI_T1	VI_T2	VI_T3	VI_T4	Yield_stage1	Yield_E	Yield_T	Yield_L	Yield_D	
Stage-II	VI_T1	1	 	 	 	 	 	 	 	 	
VI_T2	0.544**	1	 	 	 	 	 	 	 	
VI_T3	0.309**	0.308**	1	 	 	 	 	 	 	
VI_T4	0.473**	0.388**	0.208	1	 	 	 	 	 	
Stage-I	Yield_stage1	0.044	0.082	0.077	−0.02	1	 	 	 	 	
Stage-III	Yield_E	−0.056	−0.152	0.001	0.091	0.074	1	 	 	 	
Yield_T	−0.181	−0.126	−0.048	−0.219	0.671**	0.195	1	 	 	
Yield_L	0.202	0.181	−0.049	0.19	0.174	0.223	0.346**	1	 	
Yield_D	−0.164	−0.254*	0.141	−0.091	0.323**	0.003	0.331**	0.126	1	
Notes.

Signif. codes: 0.001 ‘***’ 0.01 ‘**’ 0.05 ‘*’.

VI_T1 seedling vigour at T1

VI_T2 seedling vigour at T2

VI_T3 seedling vigour at T3

VI_T4 seedling vigour at T4

Yield_stage1 grain yield during stage-I

Yield_E grain yield under early sowing

Yield_T grain yield under timely sowing

Yield_L grain yield under late sowing

Yield_D grain yield under drought

Exposing wheat seedlings to heat and drought stresses significantly reduces the seedling vigour by affecting germination percentage, shoot length, and root length (Lu et al., 2022; Mahpara et al., 2022). We also observed a severe reduction in the seedling vigour up to 59.29 and 60.37 percent, respectively, under heat and drought stresses (Fig. 3). Seedling vigour is the result of either the effective utilization of stored food present in the seeds or the increased production of heat shock proteins (Ellis, 1987). Furthermore, we used seedling STI to identify three heat tolerant (viz., IL-50, IL-56, and IL-68) and two drought tolerant (viz., IL-42 and IL-44) ILs.

Heat and drought stresses significantly affects the grain yield and its component traits, with the susceptible genotypes showing reduced grain weight and grain number as compared to the tolerant genotypes (Blumenthal, Barlow & Wrigley, 1990; Eisenstein, 2013; Qaseem, Qureshi & Shaheen, 2019; Thistlethwaite et al., 2020). Similar observations were made in the present study, where late sowing reduced the grain yield by 72.79 percent over that of early sowing, while drought reduced the grain yield by 48.70 percent over timely sowing (Fig. S3). Heat and drought stresses severely affects grian yield during anthesis stage than in grain filling. For instance, Telfer et al. (2018) revealed a significant reduction in grain yield by 13.3 kg/ha/mm of growing season rainfall, 389 kg/ha/°C of average maximum temperature during anthesis, 302 kg/ha/°C of the number of anthesis days with >30 °C, 442 kg/ha/°C of average maximum temperature during grain filling, 161 kg/ha/°C of the number of grain filling days with >30 °C, and 182 kg/ha/°C of the number of grain filling days with >35 °C. We also observed similar results, where the number of days with more than 30 °C and 35 °C had maximum impact during anthesis stage (with a yield reduction of 71.98 and 611.85 gm/plot/day, respectively) than in grain filling stage (with a yield reduction of 50.99 and 53.21gm/plot/day, respectively) (Fig. 6). This might be owing to reduced pollen viability under heat stress, which indirectly reduced the grain number per spike (Ferris et al., 1998). Similarly, we observed a 48.70 percent yield reduction under drought stress, with an average 9.88 gm/plot/cm irrigation water (Fig. 6). This is owing to increased osmotic potential coupled with reduced sink strength under drought stress (Qaseem, Qureshi & Shaheen, 2019). Furthermore, we also observed a larger effect of growing season temperature during the grain filling stage than that during the anthesis stage. This might be owing to the heat indeuced leaf senescence and reduced grain filling duration (Talukder, McDonald & Gill, 2013; Qaseem, Qureshi & Shaheen, 2019; Tanin et al., 2022).

A significant GEI for gain yield and component traits indicates the differential response of genotypes to the stress treatments (Setimela et al., 2015; Telfer et al., 2018). We also observed significant (P-value < 0.05) genotype × heat and genotype × drought interactions, which unveiled the difference in the performance of genotypes under heat and drought stress, respectively (Table 2). In addition, due to the presence of significant genotype × year interactions, several studies emphasize the importance of evaluating heat and drought tolerance over multiple years (Telfer et al., 2018; Tanin et al., 2022). Significant genotype × year interactions observed in the present study for both heat and drought stresses indicate the response of grain yield over the years differed among genotypes. This is supposed to be due to the differences in the magnitude of climate covariates associated with heat and drought stress. For instance, compared to 2020–21, the temperature covariates were very high during 2021–22, specifically during anthesis and grain filling stages (Fig. 6). This reduced the grain yield by 13.08, 47.61, and 24.18 percent, respectively, under early, timely, and late sowing in 2021–22 (Fig. S3). Similarly, the amount of rainfall in 2020–21 (19.3 mm) during anthesis and grain filling stages (specifically, after 10th week on the standard meteorological week) was significantly higher than that in 2021–22 (0 mm), which reduced the grain yield by 22.23 percent (Fig. 6). These results suggest the presence of a negative correlation between temperature and rainfall and also reveal the simultaneous occurrence of heat and drought stress under field conditions (Shah & Paulsen, 2003; Telfer et al., 2018).

Pooled ANOVA helps in estimating the GEI and describe the main treatment effects (Hakim et al., 2021). However, it failed to explain the prevailing GEI. There are several univariate and multivariate stability models to precisely explain the GEI identified in the multi-environment datasets (Eberhart & Russell, 1966; Gauch, 1992; Yan & Tinker, 2006; Olivoto et al., 2019). Multivariate models such as GGE-biplot, AMMI model, and WAASB are among the most powerful approaches for explaining prevailing GEI. GGE-biplot evolved as a most comprehensive stability model, where specific questions related to genotype and environment can be addressed graphically (Yan & Tinker, 2006). The AMMI model splits the total GEI variation into IPCs to explain the GEI (Gauch, 1992), whereas the WAASB estimates the weighted averages of absolute scores from the singular value decomposition (SVD) of the BLUP matrix for graphical visualization of GEI effects generated by linear mixed models (Olivoto et al., 2019). Utilizing different stability models for the same multi-environment datasets will assign different ranks to the same genotype. This is owing to the statistical constraints associated with each of the models. For instance, initial PCs of the AMMI model (IPCA1) and GGE-biplot (PC1 and PC2) will explain only a part of the total GEI variation and hence assign wrong ranks to the genotypes by omitting most of the GEI variation (Olivoto et al., 2019). Similar observations were made in the present study, where initial PCs of the AMMI model and GGE-biplot explained only 57.76 and 75.76 percent of the total GEI variation, respectively, and identified IL-98 and IL-259 as the most stable genotypes. However, the rank of the genotypes will be changed if we include all the significant IPCs. This can be achieved by estimating the WAASB scores suggested by Olivoto et al. (2019). By giving complete weightage to genotype stability, WAASB estimates identified IL-98 as the most stable genotype. In practice, breeders or farmers will not sacrifice yield for the sake of finding stable performing genotypes. Therefore, it is necessary to give high weightage to the grain yield while doing stability analysis. In this direction, Olivoto et al. (2019) suggested calculating the superiority index (also called WAASBY) by giving varied weightages to the grain yield and stability during multi-environment data analysis. In this study, we identified IL-47, IL-51, and IL-259 as the high-performing stable ILs for heat and drought tolerance based on WAASBY scores calculated by giving a weightage of 65 percent to grain yield and 35 percent to stability.

The identified ILs with wider and specific adaptability can be used as pre-breeding materials to transfer heat and drought responsive genes into the cultivar background. However, transferring such genomic regions through conventional approaches is tedious and time-consuming. Therefore, future studies must focus on identifying candidate genomic regions associated with heat and drought stress tolerance in these ILs. Once such genomic regions are identified, they can be employed in marker-assisted breeding to heat and drought resilient wheat cultivars

Comparing the effectiveness of different stages

For comparing different screening techniques, we used seedling vigour from stage-II and seed yield from stage-I and stage-III. Studies reported a lack of correlation between screening methods involving heat and drought stress at seedling stage and anthesis and grain filling stage (Poorter et al., 2016). Similarly, no significant correlation was observed between seedling vigour under growth chambers and seed yield under field conditions. Seed yield under field screening is a complex interaction of several climatic factors (Hede et al., 1999). Therefore, the genotypes showing higher seedling vigour did not showed higher seed yield in field experiments. Therefore, the advanced screening methods, such as heat chambers or moving shelters, need to be included in future studies.

Conclusion

Preliminary evaluation of 369 Ae. tauschii ILs for agronomic traits helped in selecting 59 agronomically superior ILs. Evaluating selected ILs under heat and drought stress significantly reduced the seedling vigour and seed yield. The identified ILs with wider and specific adaptability can be used as pre-breeding materials to transfer heat and drought responsive genes into the cultivar background. However, transferring such genomic regions through conventional approaches is tedious and time-consuming. Therefore, future studies must focus on identifying candidate genomic regions associated with heat and drought stress tolerance in these ILs. Once such genomic regions are identified, they can be employed in marker-assisted breeding to develop heat and drought resilient wheat cultivars.

Supplemental Information

Supplemental Information 1 Raw data

Supplemental Information 2 Weekly weather data on

(a-c) mean maximum and minimum temperatures (° C), total rainfall (mm), mean sun-shine hours (hours/day), and mean relative humidity (%) for 2019-20 (a), for 2020-21 (b), for 2021-22 (d); (d) mean temperature for 2019-20, 2020-21, and 2021-22

Supplemental Information 3 Statistical analysis for agronomic traits during stage-I in the 59 selected introgression lines (ILs)

(a-e) linear regression analysis of grain yield with spike length (a), spikelet number (b), plant height (c), tiller number (d), 1000 seed weight (e); (f) Pearson’s correlation between different agronomic traits; (g) principal component analysis for agronomic traits.

Supplemental Information 4 Effect of heat and drought stress on

(a) days to heading; (b) spikelet number; (c) spike length; (d) plant height; (e) tiller number; (f) days to maturity; and (g) grain yield (gm/plot) over 2020-21 and 2021-22. E1: early sowing in 2020-21; E2: early sowing in 2021-22; E3: timely sowing in 2020-21; E4: timely sowing in 2021-22; E5: late sowing in 2020-21; E6: late sowing in 2021-22; E7: rainfed sowing in 2020-21; E8: rainfed sowing in 2021-22.

Supplemental Information 5 Ranking of introgression lines based on number of IPCA

Supplemental Information 6 Ranking of introgression lines based on different weightages given to stability (WAASB scores) (0 to 100) and grain yield (100 to 0)

Supplemental Information 7 List of germplasm lines (380 + 2 checks) used during stage-I

Supplemental Information 8 List of germplasm lines (including 59 selected Aegilops tauschii derived introgression lines) used during stage-II and stage-III for evaluation under heat and drought stress tolerance

Supplemental Information 9 Analysis of variance (ANOVA; F-value) for agronomic traits of 384 genotypes screened during stage-1

Supplemental Information 10 Analysis of variance (ANOVA) for seedling vigour under different growing conditions in stage-II

Supplemental Information 11 Single and pooled analysis of variance (ANOVA; mean sum of square) for agronomic traits in stage-III under different environments (E1 to E8)

Supplemental Information 12 Effect of different climate covariates on grain yield in the Aegilops tauschii derived introgression lines (ILs)

Supplemental Information 13 Ranking of genotypes and environments based on yield performance (OrResp), stability (OrWAASB), PC1 (OrPC1), and combined yield and stability (OrWAASBY)

Supplemental Information 14 Adding an author

The authors are thankful to the Punjab Agriculture University, Ludhiana for providing facilities and germplasm resources for conducting research.

Abbreviations

ILs Introgression lines

GGE biplot genotypic main effect plus genotype × environment interaction

AMMI Additive Main Effects and Multiplicative Interaction

WAAS weighted average of absolute scores

BLUP best linear unbiased predictions

Additional Information and Declarations

Competing Interests

Author Contributions

Data Availability

Diaa Abd El-Moneim is an Academic Editor for PeerJ.

Santosh Gudi conceived and designed the experiments, performed the experiments, analyzed the data, prepared figures and/or tables, authored or reviewed drafts of the article, and approved the final draft.

Mohit Jain performed the experiments, prepared figures and/or tables, and approved the final draft.

Satinder Singh analyzed the data, prepared figures and/or tables, and approved the final draft.

Satinder Kaur conceived and designed the experiments, authored or reviewed drafts of the article, and approved the final draft.

Puja Srivastava conceived and designed the experiments, authored or reviewed drafts of the article, and approved the final draft.

Gurvinder Singh Mavi conceived and designed the experiments, authored or reviewed drafts of the article, and approved the final draft.

Parveen Chhuneja conceived and designed the experiments, authored or reviewed drafts of the article, and approved the final draft.

Virinder Singh Sohu conceived and designed the experiments, authored or reviewed drafts of the article, and approved the final draft.

Fatmah A. Safhi analyzed the data, authored or reviewed drafts of the article, and approved the final draft.

Diaa Abd El-Moneim analyzed the data, authored or reviewed drafts of the article, and approved the final draft.

Achla Sharma conceived and designed the experiments, authored or reviewed drafts of the article, and approved the final draft.

The following information was supplied regarding data availability:

The raw measurements are available in the Supplementary Files.

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
