# Peer review of "Stress adaptive plasticity from Aegilops tauschii introgression lines improves drought and heat stress tolerance in bread wheat (Triticum aestivum L.)"

_PeerJ, doi:10.7717/peerj.17528_

## Round 0.1 · original submission · Minor Revisions

Your manuscript deals with an interesting topic, and although it is very good scientifically, methodologically and in terms of written language, it needs minor corrections.

·

Basic reporting

1. I found some minor grammatical errors in the MS. It needs to be corrected.
2. Literature references, sufficient field background/ context is provided.
3. Raw data shared but some supplementary tables of S2, S3, S4, and S5 are missing.
4. It contains relevant results to hypotheses.
5. Author should be following the journal guidelines to make the abstract, no need to give key messages as per journal guidelines.
6. Throughout the manuscript many places missed the figure and table number and, author should be checked very carefully.
7. L64 delete “along with parental lines and check varieties”
8. L66 delete “the” and “(along with several climate covariates)”
9. L80 provide estimated new data of 2023/2024.
10. L82-91 cite the reference
11. L138-139 delete “The ABLs are the improved versions of DBW17 (i.e., BWL3279), PBW343 (i.e., BWL3531), and HD2967 (i.e., BWL4444) with different rust resistance genes”.
12. L104 change “heat wave” “heat stress”.
13. I suggest citing the recent reference (specially from 2022 onwards).
14. Reference listings need to be checked properly and formatted according to the journal guidelines.

Experimental design

1. Research question well defined, relevant and meaningful.
2. Methodology of multiple stage evaluation of ILs are described with sufficient detail and appropriate as per the objectives.

Validity of the findings

1. L233-235: seed yield data of both checks do not match with raw data table, should be check carefully.
2. L419-424 - Mentioned table or figure number.
3. L237 -Provide test weight of 1000 grains of check varieties of PBW725 and HD3086, which is missing in raw data table.
4. L250 - Mentioned data table of seed vigour of all four treatments (heat + drought)
5. L257 - Give data table of heat tolerance ILs scored lines with respect to heat stress. from figure no.4, it's not possible to select the tolerance ILs.
6. L260 - lack of drought scoring data of all ILs line along with check varieties.
7. L261-262 - Table 3 is not sufficient to explain the tolerance level of few selected ILs, provide data to explain the statement.
8. L283 - lack of information of ILs yield data in table 5, provide data of all ILs based on superiors ILs were selected.
9. L301 to 302 - lack of data information at anthesis and grain filling stages, provide all data.
10. L305 to 307 - provide data of ILs with respect to drought.

·

Basic reporting

It is recommended to review it again due to spelling errors.
The presentation of the statistical analysis and data evaluation tables is understandable and descriptive.

Experimental design

• It is important that the material consists of Aegilops tauschii derived introgression lines.

Validity of the findings

The findings and discussion are supported by the literature but are too long. The length of the study is similar to the thesis study.

Additional comments

Drought and heat are environmental stressors that threaten crop production worldwide. It is important to grow plant species and varieties that are tolerant in terms of the physiological mechanisms developed against these stress factors. More than 50% of wheat-growing areas, of which India has the largest share, are affected by periodic drought. Although drought affects wheat development in all developmental periods, the plant is much more sensitive during anthesis and grain filling periods.

·

Basic reporting

General comment
Overall, this manuscript presents a comprehensive study on the evaluation of Aegilops tauschii-derived introgression lines (ILs) for heat and drought tolerance in wheat. The study is very interesting and well-designed, and the findings may give significant resources for research and may aid in deciphering the Salicaceae's complicated evolutionary architecture. I have some minor comments which could be addressed to improve the manuscript and increase the chance of publication.
Here are some specific comments and suggestions for improvement:
1. Clarity and Organization: The manuscript is well-structured and organized, with a clear delineation of the objectives, methods, results, and conclusions. However, some sections could benefit from further clarification or elaboration, particularly in explaining the rationale behind certain experimental approaches or data interpretation.
2. Methodological Rigor: The methodology appears robust, with appropriate experimental designs and statistical analyses. However, more details could be provided regarding specific protocols and procedures, particularly in the seedling evaluation and field trials sections, to ensure reproducibility by other researchers.
3. Results: Additional discussion on the implications of the findings in the context of previous research or theoretical frameworks could enhance the manuscript's scientific significance.
4. Significance and Novelty: The manuscript addresses an important issue in agriculture - improving stress tolerance in wheat - and utilizes novel genetic resources from Aegilops tauschii. Emphasizing the novelty and potential impact of the findings could strengthen the manuscript's contribution to the field.
5. Language and Style: Overall, the language is clear and concise. : Additionally, there are instances where terminology could be clarified or jargon reduced to improve accessibility to a broader audience, particularly in describing complex genetic concepts or experimental techniques.
6. Figures and Tables: The figures and tables provided are informative and well-designed. : Additionally, ensuring consistency in labeling and clarity in presentation would enhance their effectiveness in conveying key results and trends.
7. Future Directions: It would be beneficial to include a brief discussion on future research directions or potential applications of the findings, particularly regarding further genetic studies or breeding efforts aimed at developing stress-tolerant wheat varieties.
# Also check all references in the text and reference list carefully some are missing from the list or not cited in the text.
Overall, the manuscript presents valuable research on improving stress tolerance in wheat through the utilization of introgression lines from Aegilops tauschii. Addressing the above points could strengthen the manuscript and increase its impact in the field of agricultural genetics and breeding

Experimental design

Methodological Rigor: The methodology appears robust, with appropriate experimental designs and statistical analyses. However, more details could be provided regarding specific protocols and procedures, particularly in the seedling evaluation and field trials sections, to ensure reproducibility by other researchers.

Validity of the findings

Significance and Novelty: The manuscript addresses an important issue in agriculture - improving stress tolerance in wheat - and utilizes novel genetic resources from Aegilops tauschii. Emphasizing the novelty and potential impact of the findings could strengthen the manuscript's contribution to the field.

Figures and Tables: The figures and tables provided are informative and well-designed. : Additionally, ensuring consistency in labeling and clarity in presentation would enhance their effectiveness in conveying key results and trends.

Future Directions: It would be beneficial to include a brief discussion on future research directions or potential applications of the findings, particularly regarding further genetic studies or breeding efforts aimed at developing stress-tolerant wheat varieties.

Additional comments

Language and Style: Overall, the language is clear and concise. : Additionally, there are instances where terminology could be clarified or jargon reduced to improve accessibility to a broader audience, particularly in describing complex genetic concepts or experimental techniques.

---

## Round 0.2 · accepted · Accept

I am pleased to inform you that your manuscript has met all corrections and is suitable for acceptance.

·

Basic reporting

Suitable for printing.

Experimental design

Suitable for printing.

Validity of the findings

Suitable for printing.

·

Basic reporting

The manuscript has been well revised by the authors and its current version is appear nicely the importance of work which might be interesting to the readers, researchers, and scientist of the PeerJ journal as well as meets the journal standards.

Experimental design

This research study used a good experimental design that expressed well in the materials and methods section of the manuscript, and it helped to generate a good finding/s from the investigation in addition to supporting the rational of the presented study.

Validity of the findings

The key results obtained from this research study have great validity in the context of breeding climatic resilient wheat cultivars as an urgent breeding goal for sustainable wheat production.

Additional comments

No more comments have to be added.